# Molecular Dynamics Study of the Green Solvent Polyethylene Glycol with Water Impurities

**DOI:** 10.3390/molecules29092070

**Published:** 2024-04-30

**Authors:** Markus M. Hoffmann, Matthew D. Too, Nathaniel A. Paddock, Robin Horstmann, Sebastian Kloth, Michael Vogel, Gerd Buntkowsky

**Affiliations:** 1Department of Chemistry and Biochemistry, State University of New York Brockport, Brockport, NY 14420, USA; 2Institute of Condensed Matter Physics, Technical University Darmstadt, Hochschulstraße 6, 64289 Darmstadt, Germanymichael.vogel@pkm.tu-darmstadt.de (M.V.); 3Institute of Physical Chemistry, Technical University Darmstadt, Alarich-Weiss-Straße 8, 64287 Darmstadt, Germany

**Keywords:** polyethylene glycol, ethylene glycol oligomers, water impurity, hydrogen bonding, radial distribution functions, density, self diffusion, viscosity

## Abstract

Polyethylene glycol (PEG) is one of the environmentally benign solvent options for green chemistry. It readily absorbs water when exposed to the atmosphere. The Molecular Dynamics (MD) simulations of PEG200, a commercial mixture of low molecular weight polyethyelene glycol oligomers, as well as di-, tetra-, and hexaethylene glycol are presented to study the effect of added water impurities up to a weight fraction of 0.020, which covers the typical range of water impurities due to water absorption from the atmosphere. Each system was simulated a total of four times using different combinations of two force fields for the water (SPC/E and TIP4P/2005) and two force fields for the PEG and oligomer (OPLS-AA and modified OPLS-AA). The observed trends in the effects of water addition were qualitatively quite robust with respect to these force field combinations and showed that the water does not aggregate but forms hydrogen bonds at most between two water molecules. In general, the added water causes overall either no or very small and nuanced effects in the simulation results. Specifically, the obtained water RDFs are mostly identical regardless of the water content. The added water reduces oligomer hydrogen bonding interactions overall as it competes and forms hydrogen bonds with the oligomers. The loss of intramolecular oligomer hydrogen bonding is in part compensated by oligomers switching from inter- to intramolecular hydrogen bonding. The interplay of the competing hydrogen bonding interactions leads to the presence of shallow extrema with respect to the water weight fraction dependencies for densities, viscosities, and self-diffusion coefficients, in contrast to experimental measurements, which show monotonous dependencies. However, these trends are very small in magnitude and thus confirm the experimentally observed insensitivity of these physical properties to the presence of water impurities.

## 1. Introduction

Polyethylene glycol (PEG, H-[O-CH_2_-CH_2_]_n_-OH) has been explored as an alternative solvent in chemical synthesis for about three decades. It is an attractive medium for chemistry because PEG is non-toxic, possesses a low vapor pressure, which reduces exposure through inhalation, and is biodegradable [1]. First reviews on the initial successes of PEG in chemical synthesis appeared in 2005 [2,3]. More recent reviews [4,5,6,7] demonstrate that PEG is not only an environmentally friendly chemical solvent but also capable of dissolving a wide range of substances [8] including, to some extent, several mineral salts [9]. This explains why PEG is a particularly effective solvent for one-pot multi-component synthesis procedures that result in quite complex chemical structures without the need of repeated product isolation and purification [5]. Another attractive feature of PEG is that it is widely available and relatively inexpensive because PEG is an important industrial chemical widely used as an additive in the personal and health care industries [10,11,12] and produced annually on the order of 500,000 tons [13]. Industrially produced liquid PEG is a polydisperse mixture of low molar weight ethylene glycol oligomers, henceforth referred to as oligomers, where the product name reflects the average molar mass such as, for example, 200 g·mol^−1^ for PEG200.

Despite these many interesting characteristics of PEG and its successful use in chemical synthesis, a physicochemical understanding of PEG as a solvent is lacking and many physical properties are not well established and/or are available only over a limited range of temperatures as recently pointed out [4]. To address this lack of knowledge and further aid the ongoing efforts in utilizing PEG as a green solvent, we have recently conducted experimental [14,15] and Molecular Dynamics (MD) simulation studies [16] on PEG and its oligomer constituents. The experimental studies comprised the measurements of density, viscosity, and self-diffusion coefficients, which carefully accounted for the potential impact of the present water impurities on the measurements. Water is a common impurity in many chemicals as they may absorb water when exposed to the open atmosphere. While working under inert atmosphere using, for example, Schlenk techniques or inert gas chambers commonly referred to as gloveboxes are an appropriate remedy, these remedies require extra equipment and time resources and may be costly to implement in an industrial setting. Therefore, it is helpful to understand the effect of water impurities on the chemicals of interest to discern to what extent measures to prevent their exposure to open atmosphere are necessary for a given application. Surprisingly, it was found that water impurities up to ~0.02 mass fraction, which is ~0.15 in mol fraction, hardly change densities, viscosities, and self-diffusion coefficients of oligomers up to nonaethylene glycol as well as PEG200 and PEG400. By comparison, other solvents show stronger effects to the presence of water impurities. For example, the viscosity of some ionic liquids reduces notably even with very small amounts of water impurities [17,18,19].

The objective of this study was to better understand why the presence of water impurities in PEG does not cause any significant changes in its physical properties. As initial hypotheses we considered two possibilities: (a) the water might aggregate in pools of water and thus does not interfere much in the intermolecular interactions of PEG and (b) there are canceling effects that the water causes in the intermolecular interactions of PEG. MD simulations are particularly suitable to find answers to such qualitative molecular-level questions. This study builds on our prior MD simulation work on PEG200 where available force fields reported in the literature were evaluated [16]. Unfortunately, none of the tested force fields reproduced well experimental values for the density, viscosity, and self-diffusion coefficient. The All-Atom Optimized Potential for Liquid Simulations (OPLS/AA, henceforth referred to as “OPLS”) force field [20] appeared to display the best compromise performance of reproducing the experimental measurements of all three of these physical properties. Adjustments of the OPLS force field were also attempted. It was found that decreasing the polarity of the hydroxyl end group and restraining less the rotation of the terminal C-C bond, as illustrated in Figure 1, led to improved but still imperfect agreement with the experimental values. Given the lack of a force field for the PEG oligomer that would produce simulated properties that are in excellent agreement with experimental values, the approach for this study was chosen to simulate PEG200 as well as three of its individual oligomer components, namely di-, tetra-, and hexaethylene glycol, using both the unmodified and the modified OPLS forcefield. This allows for checking if the observed trends are sensitive to the choice of the PEG200 force field. In this regard, it is important to note the finding of the prior MD study that the modification of the force field reduced hydrogen bonding overall in the PEG200 with a concurrent shift towards intramolecular hydrogen bonding. As is well known, water is a hydrogen acceptor and donor for hydrogen bonding. Its presence significantly increases the complexity of these present hydrogen bonding interactions involving the hydroxy (“OH”) and ether (“OE”) moieties of the oligomers, as summarized in Figure 1. Thus, comparing results from the simulations with the modified and unmodified OPLS force field might reveal additional insights into how water interferes with the intermolecular interactions in PEG.

Likewise, two different water force fields were used as well, which are the three-point SPC/E model [21] and the four-point TIP4P/2005 model [22]. Together, this amounts to four simulations, each with a different combination of the PEG and water force field, to discern to what extent the observed trends in the simulation results for each PEG–water system are dependent on the choice of the force field combination. Since the experimental investigations [14,15] found the same water independence behavior at each of the investigated temperatures (298 K–358 K), the MD investigations were performed at one temperature. The chosen temperature of 328 K matches that of our prior MD simulation study [16] and thus allows the inclusion of these prior simulation results in this study for further comparisons.

The organization of this report is as follows: Details on carrying out the MD simulations and analyzing the obtained trajectories are provided in the Methods section. The Results and Discussion section begins with an overview of the obtained results and then carefully inspects the radial distribution functions (RDFs) involving water as these set the basis to interpret the trends in the hydrogen bonding interactions upon the addition of water. The improved understanding of the hydrogen bonding interaction in turn helps to explain the trends in the physical properties upon the addition of water that are discussed next. Finally, the main insights are summarized in the Conclusions section, which also highlights their significance for applying PEG as a solvent.

## 2. Results and Discussion

An exemplary MD simulation snapshot is shown in Figure 2 for the case of PEG200 with water mass fraction, *w_water_* = 0.020. Even at this highest level of water content there is no aggregation of the water molecules observable. Instead, the water is found to interact with typically not more than one other water molecule. More prevalent are the interactions of the water with the PEG components, where the water is observed to maximize these interactions by being as close to as many OH and OE moieties as possible, as exemplary shown in Figure 1b. Clearly, one of the two hypotheses that the water may form clusters in the PEG can be ruled out. As will be shown in this section, the obtained combined simulation results do confirm that the added water does not significantly impact the physical properties of the PEG200. The observed changes with the increasing water content are very small, hardly above the respective uncertainties (not shown in the respective graphs to avoid clutter). However, the observed trends are oftentimes persistent across each set of the four simulations. Therefore, they warrant close inspection, as is done in this section, to gain insights into the competing effects of the present water on the intermolecular interactions. We begin with a close inspection of the RDFs.

### 2.1. Radial Distribution Functions

Figure 3 shows the three sets of the following RDFs: water oxygen with water oxygen (*g*_water,water_) in Figure 3a and in Figure 3b water oxygen with hydroxy oxygen (*g*_water,OH_, black) and water oxygen with ether oxygen (*g*_water,OE_, red) of diethylene glycol, each from different water mass fractions, *w_water_*, of 0.001, 0.005, 0.010, and 0.020. Aside from the different noise levels, the respective RDFs are completely indistinguishable. The same water content independence has also been observed for the tetraethylene glycol, hexaethylene glycol, and PEG200, regardless of which force fields were used. The water content independence of the RDFs suggests that the way water interacts with the oligomers and PEG200 does not change with the increasing water content up to *w_water_* = 0.020. Given the water content independence, the following graphical data on the RDFs presented in this subsection are all obtained for *w_water_* = 0.020, where there is the least noise present.

Next, we inspect the effect of different force-fields on the water–water, water-OH, and water-OE RDFs, exemplary for hexaethylene glycol. As can be seen in Figure 4, the RDFs are of the same shape independently if the TIP4P/2005 or the SPC/E forcefield is used for the water. The only noticeable difference is a slightly smaller first peak near 0.25 nm for the water–water RDF for TIP4P/2005.

When comparing the effects between the OPLS and the modified OPLS force fields, some differences are observed for the water–water and the water-OH RDFs in Figure 5, while the water-OE RDFs are nearly identical. Specifically, in the case of the modified OPLS force field, the first peak slightly below 0.3 nm is larger in the water–water RDF but smaller in the water-OH RDF. In addition, the small feature near 0.5 nm present in the water–water RDF obtained from the OPLS force field is absent for the water–water RDF obtained from the modified OPLS force field. These relatively minor differences diminish in the respective force fields for di- and tetraethylene glycol (Appendix A). Apparently, the lower dihedral barrier in the modified OPLS forcefield does not affect the oligomer–water interactions, but the additional lowering of the polarity of the hydroxy group, which is only applied for the modified forcefield of the hexaethylene glycol but not for di- and tetraethylene glycol, does.

Next, we inspect exemplary for tetraethylene glycol if there are any differences in the RDFs of the individual oligomers when they are obtained from the simulations of the water with the single oligomer component or from the simulations of the PEG200, i.e., in a mixture with other oligomers. Figure 6 clearly shows that both the water-OH and water-OE RDFs are nearly identical in this comparison. (For clarity, we note that the water–water RDFs cannot be compared because of the presence of all oligomers in PEG200.) This indicates that the interactions of water with each oligomer are not significantly influenced by the presence of other oligomers, and the water—PEG200 RDFs can be predicted from the water-single oligomer component RDFs. In fact, we checked that the water-OH and water-OE RDFs for the PEG200 can be understood as the mole fraction-weighted contributions of these RDFs from each oligomer component and confirmed that the calculated and actual RDFs are indeed identical as expected (Appendix A). The same holds true for neat PEG200 without the presence of water. This suggests that the physical properties of the PEG200 should generally be calculable from the mole fraction-weighted contribution of each neat component. This has indeed been observed in an experimental study of the molar volume, viscosity, and self-diffusion coefficient [14]. Furthermore, given the observed robustness of the RDFs with respect to changes in the water content, water force field, and oligomer force field, only very small variations should be expected for these and other physicochemical properties upon the addition of water. This will be inspected in Section 2.3.

The task remaining for this subsection is to provide a molecular level interpretation of the RDFs. Clearly, water is hydrogen bonding to other water molecules as well as to the hydroxy and ether functionalities of the oligomers as schematically summarized in Figure 1. The O-O atom-to-atom distance is approximately the same in each case, which is reflected by the first peak at 0.280–0.285 nm present in all RDFs in Figure 3, Figure 4, Figure 5 and Figure 6. The height of this first peak is similarly high in all shown RDFs except for the water–diethylene glycol ether RDF in Figure 3b, where this peak is very small. This can readily be explained by the presence of just one ether functionality in the diethylene glycol. As shown in Figure 2b, the oligomers may configure themselves to allow for the maximum closeness of their ether and hydroxyl oxygen atoms to water. In this particular snapshot of Figure 2b, the water oxygen atom is separated by slightly less than 0.3 nm to one hydroxyl atom and three ether oxygen atoms, which is consistent with the first peak position in the RDFs. The distance to the remaining two ether atoms of the hexaethylene glycol in Figure 2b is between 0.4 and 0.5 nm, which is consistent with the position of the second peak in the water-OE RDFs in Figure 3, Figure 4, Figure 5 and Figure 6. Likewise, when the water molecule is hydrogen bonded to an ether oxygen, the distance of the oligomer hydroxy group may also be 0.4 nm–0.5 nm away from the water oxygen atom, hence the similar second peak in the water-OH RDFs in Figure 3, Figure 4, Figure 5 and Figure 6. There is an additional contribution to this second peak in the water-OH RDF that arises from the scenario when the oligomer hydroxyl group is engaged in hydrogen bonding not only to water but also another oligomer hydroxyl group. It is also possible that two water molecules hydrogen bond to the same hydroxyl group, although this was only occasionally observed when expecting the simulation trajectory. Nevertheless, the scenario of the two water molecules forming hydrogen bonds to the same hydroxy group may explain the small peak in the water–water RDFs in Figure 3a and Figure 4a near 0.5 nm. This peak is absent in Figure 4a for the RDF obtained with the modified OPLS force field. This can be explained by the prior observation that the modifications to the OPLS forcefield result in an overall reduction in hydrogen bonding and a shift from inter- towards intramolecular hydrogen bonding [16]. A third peak near 0.7 nm is observed in all of the three RDFs, water–water, water-OH, and water-OE (except for diethylene glycol in Figure 3b) in Figure 3, Figure 4, Figure 5 and Figure 6. This peak arises from the scenario when the two sites of the oligomer are engaged in hydrogen bonding but such that their interaction are on opposite sides (anti) of each other. In the case of the water–water RDFs, these are two water molecules interacting in such a way with the oligomer. The additional maxima in the water–water RDFs in Figure 3, Figure 4, Figure 5 and Figure 6 are spaced in equal distance increments, which corresponds to how far the two ethylene glycol oxygen moieties to which the water hydrogen bonds are separated, i.e., by how many ethylene oxide units the two moieties are separated.

### 2.2. Hydrogen Bonding Numbers

Appendix A summarizes the number of the hydrogen bonds per number of the water molecules, *n_HB_*/*n_water_*, which are exemplary for some forcefield combinations graphed in Figure 7. Appendix A and Figure 7 are divided into four sections: water hydrogen bonded to the oligomer hydroxy groups, water hydrogen bonded to the oligomer ether groups, water hydrogen bonded to other water molecules, and the sum of these three contributions of the water hydrogen bonding. With increasing *w_water_*, *n_HB_*/*n_water_* slightly decreases for water hydrogen bonding with the hydroxy and ether groups of the oligomers, while *n_HB_*/*n_water_* for hydrogen bonding between water molecules increases. Overall, the sum of these changes mostly cancel each other so that the sums of all the water involving hydrogen bonding contributions stay nearly constant, perhaps slightly declining, with increasing *w_water_*. These observations can be explained as follows. The more water is added, the chances for water molecules to be near each other and engage in hydrogen bonding increases with a concomitant lower chance to engage in hydrogen bonding with the oligomers.

On the other hand, it is perhaps surprising that hydrogen bonding between water molecules does not increase more strongly with the addition of water. Taking a purely statistical point of view where the number of the hydrogen bonds between water molecules is related to the chance of the two water molecules to come in contact, one would expect that such contacts scale squared with the water concentration. Such thinking is based on the kinetic theory of gases [23]. Although the simulations are conducted for the condensed liquid phase, it is instructive to obtain a point of reference by evaluating *n_HB_*/*n_water_* for water–water hydrogen bonding from the kinetic theory of gases as shown in the supportive information, which results in a value of 1.68 *×* 10^−6^ for *w_water_* = 0.020. This is five orders of magnitude smaller than the corresponding values in Appendix A for several reasons. The available space for the water in the simulation box is severely limited as it is filled primarily with the PEG. This not only reduces the available volume by 1–2 orders of magnitude but also blocks the translational motion of the water. Furthermore, intermolecular interactions were neglected in this calculation, but these lead to the hydrogen bonds in the first place, which keeps the water molecules in contact and thus increases the chance of finding two water molecules hydrogen bonded to each other. Finally, the formation and breaking of the hydrogen bonds are in a dynamic equilibrium. Thus, the addition of the water leads not only to more hydrogen bonds being formed but also more hydrogen bonds being broken, and a square dependence of the number of the hydrogen bonds with respect to the water concentration cannot be expected.

It is interesting that the sum of all the water hydrogen bonding contributions is essentially independent of the oligomer size. Specifically, the values in Appendix A and Figure 7 are ranging between 2.310 and 2.810 where the highest values are observed for the PEG200 from the simulation with the unmodified OPLS forcefield and the lowest values are observed for the hexaethylene glycol when the modified OPLS forcefield is used. The latter arises mostly from the significant reduction in the water–hexaethylene glycol OH hydrogen bonding contribution that only is partially compensated by an increase in the water–hexaethylene glycol ether hydrogen bonding contribution. This behavior particular to the hexaethylene glycol is consistent with the corresponding RDFs shown in Figure 4 and can be attributed to the additional lower OH polarity that is only applied for the hexaethylene and heptaethylene glycol in the modified OPLS forcefield.

The *n_HB_*/*n_water_* values in Appendix A and Figure 7 for the diethylene glycol illustrate that the water has a much higher propensity to hydrogen bond to the hydroxy functionalities rather than the ether functionalities. Specifically, the values in Appendix A are between 0.09 and 0.15 for hydrogen bonding with the ether functional group compared to values between 2.15 and 2.52 for hydrogen bonding with the hydroxy functional group. The observation that these values are quite high, larger than the number of the hydroxy groups per the ethylene glycol of the two, indicates that the water is engaged in bridging hydrogen bonds between the ethylene glycol oligomers. Furthermore, these hydrogen bonding interactions involve the water in both ways, as hydrogen donor and as hydrogen acceptor since each water molecule can maximally donate and accept two hydrogen atoms each. By comparison, the *n_HB_*/*n_water_* values in Appendix A and Figure 7 in the cases of the tetraethylene glycol simulated with the modified OPLS force field and hexaethylene glycol are higher for hydrogen bonding to the ether than for the hydroxy functional groups. However, the number of the ether groups increases with the oligomer size, three for the tetra- and five for the hexaethylene glycol, while the number of the hydroxy groups remains constant at two. Dividing the values for hydrogen bonding to the ether oxygens in Appendix A by 3 and 5, respectively, for the tetra- and hexaethylene glycol results in both cases in values around 0.35, which is similar or smaller than half the values of the *n_HB_*/*n_water_* for the water–hydroxy hydrogen bonds (≈0.75 for tetraethylene glycol and 0.31–0.55 for hexaethylene glycol). Thus, as one might have expected, the hydrogen bonding of water to the hydroxy functionalities contributes progressively less the longer the oligomer is.

Overall, the *n_HB_*/*n_water_* values in Appendix A and Figure 7 illustrate that the water interacts strongly with the oligomers through hydrogen bonding. To compare the *n_HB_*/*n_water_* values for the total hydrogen bonding contribution with the hydrogen bonding in neat water, one needs to consider that water can be maximally engaged in four hydrogen bonding interactions, accepting and donating up to two hydrogen bonds each. With on average of about 2.6 hydrogen bonds per water molecule in Appendix A, this would be a hydrogen bonded fraction of 2.6/4 = 0.65, which is somewhat smaller than a hydrogen bonded fraction of 0.71 obtained by the interpolation of the reported temperature-dependent values for pure water [24]. The smaller hydrogen bonded fraction compared to water can be explained by the fact that a significant amount of the volume is occupied by CH_2_ groups to which water cannot hydrogen bond.

To inspect to what extent the added water competes with the oligomers in engaging in hydrogen bonding, it is helpful to inspect the oligomer OH-OH and OH-OE hydrogen bonding numbers as a function of the *w_water_* as presented in Appendix A and exemplary for some force field combinations in Figure 8. There are several interesting observations. While the oligomer–oligomer intermolecular hydrogen bonding interactions generally decrease with the increasing *w_water_*, there are notable exceptions. The numbers in Appendix A and Figure 8 are staying flat or are even increasing from *w_water_* = 0 to *w_water_* = 0.001, especially for the diethylene glycol. This indicates that oligomer–water hydrogen bonding interactions introduced by small amounts of the water bring the oligomer molecules in better alignment to each other to enable additional oligomer–oligomer intermolecular hydrogen bonding. The values in Appendix A and Figure 8 for the oligomer intramolecular hydrogen bonding interactions stay generally flat, except for the tetraethylene glycol. The independence of *w_water_* can be explained by the canceling effects the added water causes on the oligomer intramolecular hydrogen bonding interactions. On the one hand, water competes not only with intermolecular but also the intramolecular hydrogen bonding of the oligomers, which leads to a reduction in the intramolecular hydrogen bonding. On the other hand, oligomer molecules that are not intermolecular hydrogen bonded to each other due to being broken up by the water addition may be more likely to engage in intramolecular hydrogen bonding to compensate the loss of the intermolecular hydrogen bonding with other oligomers, which leads to an increase in the intramolecular hydrogen bonding. Tetraethylene glycol has been shown to have a stronger propensity to form intramolecular interactions compared to the other oligomers [16]. Hence, the effect of competing with the water is stronger for this oligomer and thus explains the decrease in the intramolecular hydrogen bonding seen in Appendix A and Figure 8. Finally, we note that the previously reported effect of the OPLS force field modifications to cause less hydrogen bonding overall with a shift towards intramolecular hydrogen bonding [16] is also observable in Appendix A and Figure 8. The choice of the water forcefield, on the other hand, has no noticeable impact on the hydrogen bond numbers, neither for the values reported in Appendix A nor the values reported in Appendix A.

### 2.3. Structure, Thermodynamics, and Dynamics

Any changes in the hydrogen bonding interactions in the oligomers upon the addition of the water should principally affect their structural properties, such as their end-to-end-distances and the radii of gyration, as well as their thermodynamic and dynamic properties such as the density, viscosity, self-diffusion coefficient, isobaric molar heat capacity, and isobaric thermal expansion coefficient. Appendix A list these properties. At first sight, the addition of the water does not appear to have any noticeable effects on any of these properties listed in Appendix A. However, upon closer look, some interesting observations can be made as exemplary shown for some force field combinations in Figure 9 and Figure 10. First off, the effect of the modifications to the OPLS force field on the properties listed in Appendix A and seen in Figure 9 and Figure 10 have been explained in detail previously [16]. The already mentioned reduction in hydrogen bonding overall with a concurrent shift towards intramolecular hydrogen bonding shortens the end-to-end distances and the radii of gyration, as well as lowers the viscosities with a concomitant increase in the self-diffusion coefficients. The densities, isobaric molar heat capacities, and isobaric thermal expansion coefficients stay relatively unaffected. These exact same trends remain observable in Appendix A in the presence of the water up to the highest investigated mass fraction of 0.020.

There are some peculiar observations for the end-to-end distances and the radii of gyration listed in Appendix A and shown in Figure 9. Both decrease from zero to 0.001 mass fraction of water for the di- and hexaethylene glycol. Even this small amount of the present water introduces sufficient intermolecular water–oligomer interactions as exemplified in the snapshot in Figure 2 to cause these decreases that are noticeably larger than the continuing decreases upon further water addition. The tetraethylene glycol behaves differently in this regard than the di- and hexaethylene glycol. The end-to-end distances and the radii of gyration increase for the simulations with the unmodified OPLS force field but decrease for the simulations with the modified OPLS forcefield. The prior observed [16] special propensity of tetraethylene glycol to form intramolecular hydrogen bonding provides an explanation for this effect because the addition of the water in this case competes with the intramolecular hydrogen bonds and replaces some of these with intermolecular hydrogen bonds with water making the end-to-end distances larger. Evidently, this additional effect dominates only for the simulations with the unmodified force field where the hydrogen bonding interactions are overall larger compared to the simulations with the modified OPLS force field. Finally, it is noteworthy that the end-to-end distances and the radii of gyration in Appendix A are most responsive for the hexaethylene glycol, which is consistent with the observations in the respective RDFs in Figure 4. They are overall also slightly more responsive to the water addition for the simulations with the unmodified OPLS force field, although still very flat with the increasing *w_water_* for the di- and tetraethylene glycol. This can be explained by less overall hydrogen bonding and a shift to intramolecular hydrogen bonding caused by the modified OPLS force field, where apparently the increases in the end-to-end bond distances and the radii of gyration from the replacement of the intermolecular hydrogen bonds with the intermolecular oligomer–water hydrogen bonds approximately cancel the decreases from the new intermolecular oligomer–water hydrogen bond interactions.

There are also some nuanced trends upon the water addition in the physical properties listed in Table 1, which for comparison also include experimental data [14,15], and are shown exemplary for some force field combinations in Figure 10.

Beginning with the densities, these have been observed experimentally to decrease by very small amounts upon the addition of water for each of the oligomers and stay constant for the PEG200. The simulated densities also show very minor changes upon the addition of water but with varying trends. For the diethylene glycol, densities first slightly decrease from 0 to 0.001 mass fraction of water but then increase slightly upon further addition of water, regardless of the force field combination. For the tetraethylene glycol, densities slightly increase throughout with increasing water content, regardless of the force field combination. For the hexaethylene glycol and for the PEG200, the trend of the densities upon water addition differs between the simulation results from the unmodified and the modified OPLS forcefield, following more that of the diethylene glycol for the unmodified forcefield but following more that of the tetraethylene glycol for the modified force field. These varied responses hint towards a fine-grained interplay between the competing factors and the system density. The steady increase in the density upon water addition observed in the tetraethylene glycol simulations suggests that the replacement of the intramolecular hydrogen bonds with intermolecular water–tetraethylene glycol hydrogen bonds is associated with a reduction in volume. The overall general increase in the simulated density for all the oligomers upon continued water addition suggests that the hydrogen bonding interactions of the oligomers with the water bring the oligomers overall closer together requiring less volume overall. The values for *n_HB_*/*n_water_* in Appendix A add up to more than two for the hydrogen bonding of water with the hydroxy and ether moieties of the oligomers indicating that the hydrogen bonding of the water with the oligomers include bridging hydrogen bonds between them, effectively acting like a glue. However, this simulated overall trend is counter to the trend in the experimental data, which suggests that the simulations might have overestimated the hydrogen bonding interactions of the water with the oligomer. Moreover, knowing that the density of pure water is below 1000 kg m^−3^ implies that the simulated densities will need to go through a maximum with further increasing the *w_water_*. One might surmise that the simulations may not capture the water behavior in so far that in reality, water preferably interacts with itself to create the tclusters of water. However, we note that also in other hydrophilic systems such as some ionic liquids, water has been observed to be spread as single or only small groups of molecules throughout the system up to the mole fractions of 0.6 [25]. More likely, the cause of this discrepancy is due to shortfalls in the forcefield of the oligomers to correctly balance the competing intricate hydrogen bonding interactions.

With respect to simulated self-diffusion coefficients, the addition of the water causes a slight decrease with increasing water mass fraction (Table 1 and Figure 10), which is more clearly observed for the simulations with the modified OPLS forcefield. The viscosities in Table 1 and Figure 10 show a concomitant slight increase with increasing water content although this trend is at least in part obscured by the larger uncertainties in obtaining the viscosities from the MD simulations (see Section 3). The inverse trend upon water addition to self-diffusion, *D*, and viscosity, *η*, is expected based on the Stokes––Einstein equation shown in Equation (1),
(1)$$D=\frac{k_{B}T}{\xi \pi \eta r}$$
where *k_B_* is the Boltzmann constant, *T* is the absolute temperature, *ξ* is a constant, and *r* is the hydrodynamic radius of the diffusing molecule.

The slight decreases in the self-diffusion coefficients and increases in the viscosities upon water addition observed for the simulations are exactly opposite to the trends of the experimental data included in Table 1. This again suggests that the MD simulations overestimate the hydrogen bonding interactions between the water and oligomers which, as noted above, results in some water molecules engaging in bridging hydrogen bonding with two oligomers.

Experimental data are unfortunately not available for the water self-diffusion coefficients. The simulated results in Table 1 and Figure 10 indicate that the water self-diffusion coefficients go through a maximum near 0.005 water mass fraction. This is also observed, albeit less strongly, for the diethylene glycol self-diffusion coefficients in Table 1. Given that neat water self-diffuses much faster than any of the neat oligomers, it can be surmised that the water self-diffusion coefficients should go through a minimum at some higher *w_water_*. The thus observed decrease in the water diffusion coefficients at intermediate water mass fractions hints that the water interacts strongly between the oligomers. Its translational motions are apparently hindered by the same barriers that the ethylene glycol oligomers experience in their translational motions because the water self-diffusion coefficients in Table 1 and Figure 10 decrease very notably with the size of the oligomer. Interestingly, the water self-diffusion coefficients in the PEG200 in Table 1 and Figure 10 are comparable if not smaller than the ones in the hexaethylene glycol even though the average molar mass of 200 g‧mol^−1^ matches more closely that of tetraethylene glycol. Apparently, the barrier of the water translational motion is limited by the larger oligomer components of the PEG200. Furthermore, we note that the simulated water self-diffusion coefficients in Table 1 are somewhat larger for the simulations with the SPC/E compared to the TIP4P/2005 force field. In this regard, the water self-diffusion coefficient is the only property that shows some significant differentiation between the two water forcefields, while all other properties listed in Appendix A show no noticeable sensitivity to the choice of the water forcefield. Given that the simulated oligomer self-diffusion coefficients are smaller than the experimental values and that the water self-diffusion is progressively slowed the longer the oligomer size is, the simulated water self-diffusion coefficients are likely also inaccurately small. Since most of the water self-diffusion constants are larger from simulations with the SPC/E forcefield compared to the TIP4P/2005 forcefield, the SPC/E forcefield appears to provide slightly more accurate results. We did not test additional water force fields given that besides the water self-diffusion coefficient all the other structural and physical properties showed water model independence. We should also point out that the OPLS and water force fields used in this study do not capture the possible effects of local electric fields, which ab initio MD simulations would capture [26]. However, these local field effects tend to increase the structural organization, which would only further increase the hydrogen bonding interactions that appear to be overestimated by the force fields tested for the oligomers of the PEG200 [16].

Finally, Table 2 shows the simulation results for the two thermodynamic properties, isobaric molar heat capacities and thermal expansion coefficients, obtained through the analysis of the fluctuations in temperature, enthalpy, and volume. The heat capacity analysis did not include the consideration of quantum effects, which is known to result in overestimates by about two-fold [27]. The two-fold overestimate is indeed confirmed by the values listed in Table 2. The addition of water lowers the system’s molar heat capacity, which is readily explained by the lower molar heat capacity of water, reported for a temperature of 328 K to be at 75.4 J‧mol^−1^‧K^−1^ [28]. The molar heat capacities listed in Table 2 show no clear differentiation between the different force field combinations employed for the simulations. Within the uncertainties in the obtained thermal expansion coefficients (see Section 3), there are no clear trends observable in Table 2 with respect to both the addition of the water as well as the combination of the force fields used.

## 3. Computational Methods

### 3.1. Simulation Details

All simulations were carried out at ambient pressure (1 bar) and 328 K using GROMACS 2020.4 [30,31] that was compiled with mixed precision. The temperature of 328 K was chosen to allow the comparison of the simulation results with prior ones on same systems without added water [16]. The same composition of the PEG200 was taken as in previous work [14,16]. The OPLS force field [20] for the PEG and ethylene glycol oligomers includes harmonic oscillator potentials for covalent bond length and bond angle vibrations with $k_{b}$ and $k_{\theta }$ and equilibrium bond lengths/angles $b_{0}$ and $\theta_{0}$, respectively, the Ryckaert–Bellemans potential with *C*_n_ torsional energy barrier coefficients for proper dihedrals, and Lennard-Jones and Coulomb potentials for the nonbonding interactions as shown in Equation (2)
(2)$$U_{total}=\sum_{bonds}^{\;}\frac{1}{2}k_{b}{\left(b-b_{o}\right)}^{2}+\sum_{angles}^{\;}\frac{1}{2}k_{\theta }{\left(\theta -\theta_{o}\right)}^{2}+\sum_{\begin{array}{c} proper\\ torsions \end{array}}^{\;}\sum_{n=0}^{5}C_{n}\cos^{n}\left(\phi -180{}^{\circ}\right)\;+\sum_{i<j}^{\;}4f_{LJ}\epsilon_{ij}\left[{\left(\frac{\sigma_{ij}}{r_{ij}}\right)}^{12}-{\left(\frac{\sigma_{ij}}{r_{ij}}\right)}^{6}\right]+\sum_{i<j}^{\;}\frac{f_{qq}}{4\pi \epsilon_{o}}\frac{q_{i}q_{j}}{r_{ij}}$$
where *q* is the partial Coulomb charges on a particular atom, *σ* and *ε*, are the Lennard-Jones (*LJ*) interaction parameters for contact distance between two atoms and well-depth, respectively, for which the Lorentz–Berthelot combination rules were applied, and *f_LJ_* and *f_qq_* are fudge factors for the Lennard-Jones and Coulomb interactions, respectively. Both fudge factors are set to 1 for all atom pairs except nonbonding interactions between 1-2 and 1-3 atom pairs, for which they are set to zero and 1-4 pairs for which they are set to 1.

Further specific details including all parameters associated with the OPLS force field are listed in completeness in the Appendix A of the prior work [16] and need not be repeated here except to summarize the adjustments of the modified OPLS forcefield which resulted in the closest agreement between simulated and experimental physical properties and are as follows (see also Figure 1): The (HO)-C-C-O dihedral potential function was reduced by ½ for di-, tri-, hexa-, and heptaethylene glycol and by ¾ for tetra- and pentaethylene glycol. In addition, the polarity of the OH bond was reduced by a net amount of 0.2 charge units for hexa- and heptaethylene glycol. The parameters for the 3-point SPC/E and the 4-point TIP4P/2005 water model are included in the GROMACS 2020.4 package and were used without any modifications.

The simulations with hexaethylene glycol consisted of 500 molecules total, while all other systems consisted of 1000 molecules total. The molecules were placed randomly with the GROMACS module “insert-molecules” into a cubic box with an initial density approximately 20% lower than the experimental density [14]. Given that the water mass fraction was kept constant for all simulations, the number of water molecules varied from system to system depending on the molar mass of the oligomer. Keeping the mass fractions rather than the mole fractions the same ensures approximately the same amount of ether functionalities present across simulated oligomer systems for a fairer comparison. The exact number composition of each system is, for clarity, summarized in Appendix A in the Appendix A section. The prepared systems were energy minimized using the steepest descent algorithm [32] where an initial maximum atom displacement of 0.01 nm was used to remove high energy contacts that might have been accidentally generated during system preparation. The “DEFLEXIBLE” option was chosen to allow the water to be treated as flexible rather than rigid molecules during this simulation step. Periodic boundary conditions were applied via the Verlet cutoff scheme [33] using a buffer tolerance of 0.005 kJ mol^−1^ ps^−1^. The cut-off distance for nonbonding electrostatics and LJ interactions were set at 1.4 nm and the long-range electrostatic interactions were accounted for using the Smooth Particle-Mesh Ewald (PME) scheme [34,35] with a grid spacing of 0.168 nm.

In the next step of the simulation protocol, the system was allowed to reach equilibrium at a constant pressure and temperature, where the simulation length was set such that the system density would converge within a tenth of the overall NPT simulation time. The simulation time steps were set at 2 fs and initial velocities were generated for each atom using a Maxwell–Boltzmann distribution. The LINCS algorithm with a fourth-order matrix expansion [36] was used to restrain all bonds involving hydrogen in every step while the bonds and angles in the water were constrained using the SETTLE algorithm [37]. Any drifting motion of the center of mass of the system was corrected every 20 fs. The treatment of the nonbonding interactions in the connection of the periodic boundary conditions was identical to that for energy minimization except that an analytic tail correction for energy and pressure for the long-range LJ potential was implemented [38]. The Bussi–Donadio–Parrinello velocity-rescaling thermostat [39] with a time constant of 1.0 ps and the Parrinello–Rahman barostat [40,41] with a time constant of 5.0 ps were used to control the temperature of 328 K and pressure of 1 bar. At least 1000 position frames were recorded for the trajectory, and at least 10,000 frames were recorded for the energies.

Starting with a position frame from NPT simulation representing the equilibrium density, the final step of the simulation protocol lets the system evolve at a constant temperature and volume long enough for the system to be ergodic and reach the diffusive regime from which the self-diffusion coefficients were extracted (100 ns, 300 ns, 800 ns, and 300 ns for di-, tetra-, hexaethylene glycol, and PEG200, respectively). Hence, pressure coupling was removed during this NVT simulation step. The energy frames were recorded every 20 fs and the positions were recorded at a frequency to result in 5000 frames. All other parameters were the same as during the previous NPT simulation.

### 3.2. Analysis

The modules available in the GROMACS package in combination with self-generated Bash or Python scripts were used to carry out the analysis of the obtained simulation trajectories, as shown in detail in the prior work [16] and briefly summarized here. Densities, isobaric heat capacities, and thermal expansion coefficients were obtained from the NPT simulation using the GROMACS module “energy” with a start time larger than the time at which the density converged. The relative standard deviations (RSDs) of these properties are 0.5%, 10%, and 20%, respectively [16]. Dynamic properties such as self-diffusion coefficients, viscosities, and hydrogen bonding numbers were obtained from the NVT simulation rather than the NPT simulation since the use of barostats in NPT is known to significantly influence dynamics [42]. The average end-to-end distances (via GROMACS module “distance”) for which the hydroxyl oxygen atoms of an H-[O-CH_2_-CH_2_]_n_-OH oligomer were chosen, average radii of gyration (via GROMACS module “gyrate”), and RDFs (via GROMACS module “rdf”) were also obtained from the NVT simulation. Both inter- and intramolecular RDFs were calculated. The intramolecular RDFs obtained from GROMACS were not normalized and were therefore rescaled to result in an integration area of 1.

The average hydrogen bond numbers over all trajectory frames were obtained using the GROMACS module “hbond” [32] in combination with the GROMACS module “analyze” [32]. The default criteria for hydrogen bond counting were selected, where acceptor–donor-hydrogen bonding triplet angles are 30° or less and acceptor–donor distances 0.35 nm or less. We did not explore other hydrogen bond definitions, for example based on energetics or topology [43], because our objective was to qualitatively compare the trends of hydrogen bonding upon the addition of water to the systems of interest. For this purpose, any hydrogen counting scheme is acceptable as long as it is used consistently. From the prior work [16], the RSDs are estimated to be 0.3%, 4%, and 12%, respectively, for intermolecular hydrogen bonds, intramolecular OH-OE hydrogen bonds, and intramolecular OH-OH hydrogen bonds.

The self-diffusion coefficients of the system components were evaluated from the average mean squared displacement of the center of mass of each component, *MSD*(*t*) (via GROMACS module “msd”). The *MSD*(*t*) functions were inspected to determine the upper and lower limits of the diffusive regime for linear regression analysis to obtain *D*(*L*) from *MSD*(*t*) = 6 *D*(*L*) *t* + *c* [42]. The obtained box size-dependent self-diffusion coefficients *D*(*L*) were adjusted to the self-diffusion coefficient at infinite box size $D_{\infty }$ using the analytic correction by Yeh and Hummer [44]
(3)$$D_{\infty }=D\left(L\right)+\frac{k_{B}T\xi }{6\pi \eta L}$$
where *k_B_* is the Boltzmann constant, *T* the temperature, *ξ* = 2.837298, and *η* the simulation viscosity. The correction typically amounted to a change of about 1–2% for diethylene glycol and water, 5% for tetraethylene glycol, and 15% for hexaethylene glycol. From the prior work, the RSD of the reported $D_{\infty }$ is less than 7% [16].

The viscosities, *η*, were obtained from the time decomposition method using the Green–Kubo (GK) integral formalism [45] in repeated fashion by splitting up the full trajectory of a single NVT simulation into multiple time blocks, as explained in detail in prior work, where the RSD for the viscosities obtained by this procedure was found to be 15% [16].

## 4. Conclusions

The simulations of the PEG200 as well as di-, tetra-, and hexaethylene glycol with added water confirm the experimental observations that the addition of the water hardly impacts their physical properties. For example, the obtained water RDFs are essentially identical regardless of the water content up to the maximum studied *w_water_* of 0.020. However, some nuanced effects on the hydrogen bonding interactions are noticed, which in turn lead to varied *w_water_* dependencies for the densities, viscosities, and self-diffusion coefficients, displaying extrema between *w_water_* = 0 and *w_water_* = 0.020 as well as at least one more extremum between *w_water_* 0.020 and neat water. These nuanced peculiar trends with the added water are counter to the experimental data, which show small *w_water_* dependencies that are monotonously increasing or decreasing functions. Although this suggests that the MD simulations might overestimate the hydrogen bonding interactions of the water with the oligomers, they do provide at least qualitatively important insights into these.

Water as a strong hydrogen bond donor and acceptor competes with the oligomer polar groups to form hydrogen bonding interactions, which thus reduces both the inter- and intramolecular hydrogen bonding within and between the oligomers. The reduction in oligomer–oligomer intermolecular hydrogen bonding increases the chance for the oligomers to engage in intramolecular hydrogen bonding. This partially offsets the loss of the intramolecular hydrogen bonds directly caused by the competition with the water. This secondary effect is particularly strong for the tetraethylene glycol, which has a very high propensity towards intramolecular hydrogen bonding.

The MD simulations also suggest that some of the added water molecules engage in hydrogen bonding with more than one oligomer, i.e., the formation of the bridging hydrogen bonds. These bridging hydrogen bonds might explain the slight decrease in self-diffusion with increasing water content in spite of the increased water–water hydrogen bonding contributions. In this respect, the simulation snapshots show that water–water hydrogen bonding is limited between two water molecules so that there is no evidence for the formation of the extended water–water hydrogen bonding networks or clusters.

Overall, the observed qualitative trends in the structure, hydrogen bonding, and bulk properties are quite robust with respect to the choice of the force field combination, where the choice of the oligomer force field (OPLS vs. modified OPLS) by and large only affects the magnitude of the trends. The simulation results suggest that the confirmed experimental observation that the addition of water hardly impacts the density, viscosity and self-diffusion of the PEG is due to the offsetting effects of (a) the water interfering with the intermolecular interactions between oligomers, (b) the secondary higher likelihood of the intramolecular hydrogen bonding of the oligomer, and (c) the occurrence of water performing bridging hydrogen bonding between the oligomers. The very practical consequence is that chemists using PEG as a solvent do not have to worry about any significant changes in the physical properties upon the absorption of water during use. This piece of information should be very useful when scaling up chemical processes that utilize PEG as a solvent. Furthermore, this molecular simulation study also confirms the prior-made observation that the physical properties of the PEG only depend on the average molar weight but not the exact composition because, in the first approximation, the physical properties can be reasonably accurately obtained from computing the mole fraction-weighted average of the property values of the oligomer components. Thus, all that is needed to predict the physical properties of the less-studied polydisperse PEG mixtures is their composition [46] and the properties of the neat oligomer components, which in fact may also be predictable based on the results of prior work comparing properties of oligomers up to nonaethylene glycol [15].

## Figures and Tables

**Figure 1 molecules-29-02070-f001:**
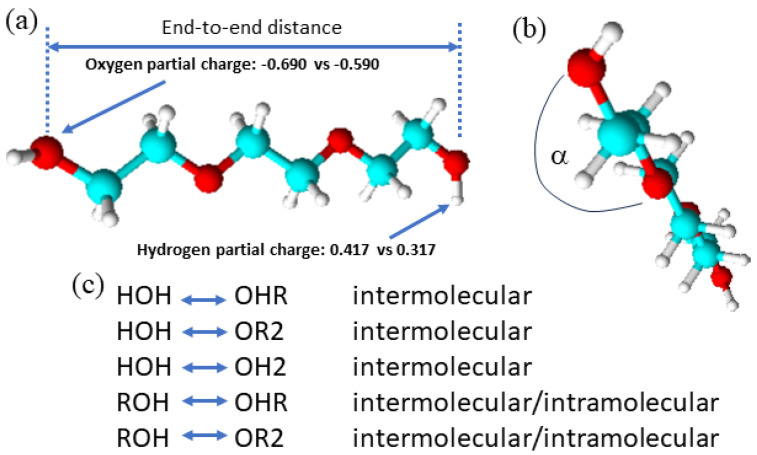
(**a**) Triethylene glycol as an example of ethylene glycol oligomers that make up PEG200, H-[O-CH_2_-CH_2_]_n_-OH, shown with the distance between the oxygen atoms of the terminal hydroxy end groups used in this report as a measure for the end-to-end distance, and the changes in the hydroxy partial charges in the modified OPLS forcefield; (**b**) an illustration of the proper dihedral angle referred to as (HO)-C-C-O in this report for which its potential energy was reduced in the modified OPLS forcefield; (**c**) a summary of possible inter- and intramolecular hydrogen bonding interactions in the studied water containing ethylene glycol oligomer mixtures.

**Figure 2 molecules-29-02070-f002:**
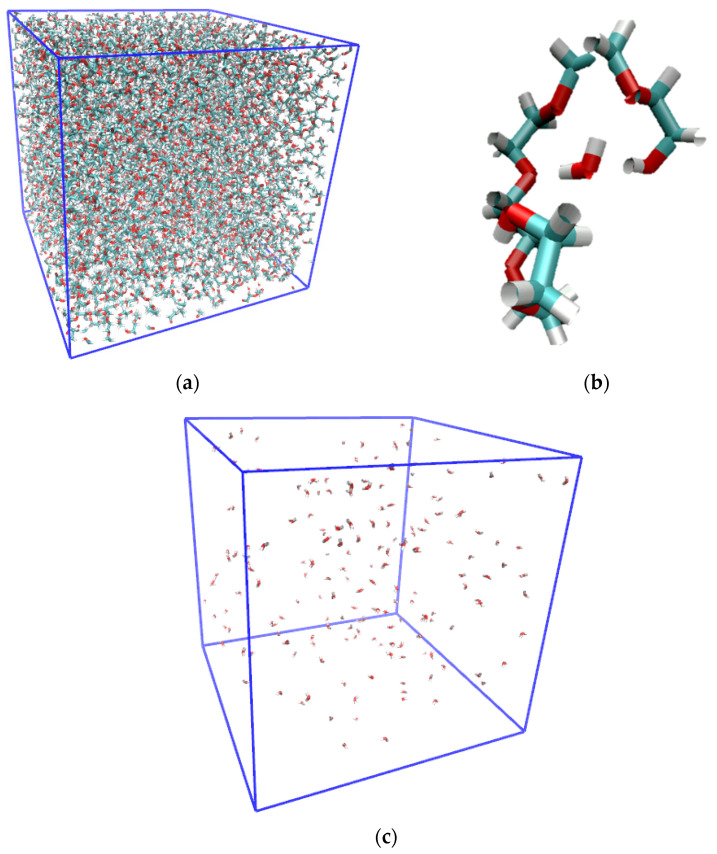
Snapshot of a MD simulation of the PEG200 with *w_water_* = 0.020 using the OPLS force field for the PEG200 and the SPC/E forcefield for the water showing (**a**) all molecules, (**b**) a water molecule interacting with hexaethylene glycol, and (**c**) just the water molecules, (oxygen = red, hydrogen = white, carbon = teal).

**Figure 3 molecules-29-02070-f003:**
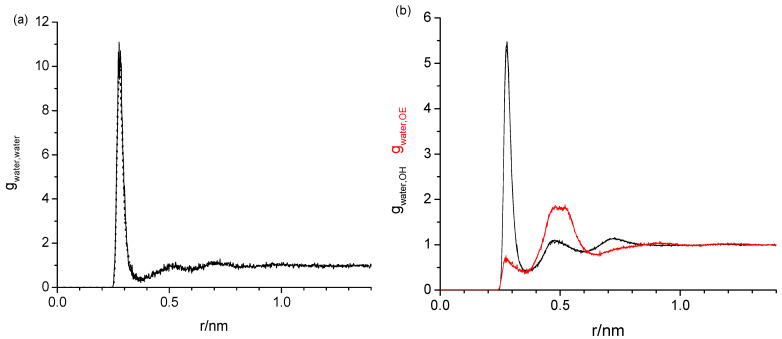
Radial distribution functions obtained with the TIP4P/2005 and OPLS forcefields of the water oxygen (**a**) with water oxygen and (**b**) with hydroxy (black) and with ether oxygen (red) of diethylene glycol, each from four different water mass fractions of 0.001 (omitted in (**a**)) to avoid clutter due to large data noise), 0.005, 0.010, and 0.020. Aside from different noise level, the respective radial distribution functions are indistinguishable.

**Figure 4 molecules-29-02070-f004:**
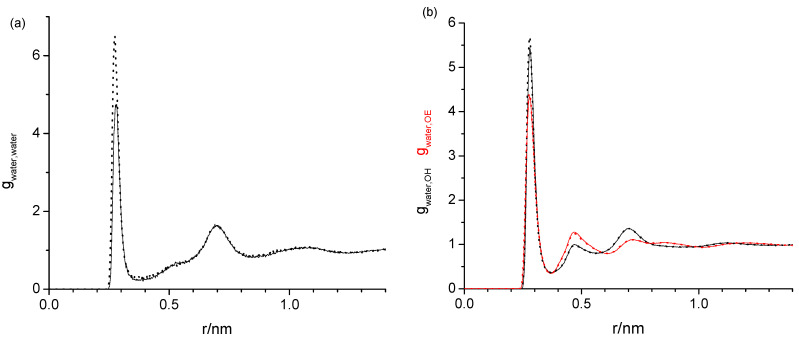
Radial distribution functions of the water oxygen with (**a**) water oxygen as well as (**b**) oligomer hydroxy oxygen (black) and ether oxygen (red) for 0.02 mass fraction of water in hexaethylene glycol obtained with the water force fields TIP4P/2005 (solid) and SPC/E (dotted) in combination with the OPLS forcefield for hexaethylene glycol. The respective radial distribution functions are essentially indistinguishable.

**Figure 5 molecules-29-02070-f005:**
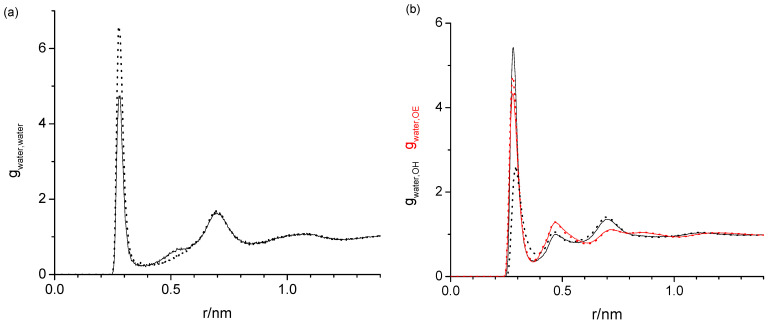
Radial distribution functions of the water oxygen with (**a**) water oxygen as well as (**b**) oligomer hydroxy oxygen (black) and ether oxygen (red) for 0.02 mass fraction of water in hexaethylene glycol obtained with the water force fields TIP4P/2005 in combination with the OPLS forcefield (solid) and the modified OPLS force field (dotted).

**Figure 6 molecules-29-02070-f006:**
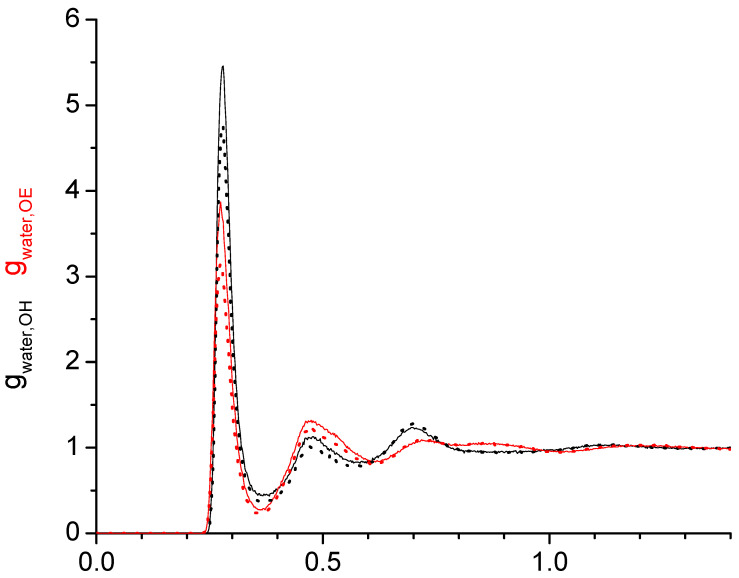
Radial distribution functions of the water oxygen–oligomer hydroxy oxygen (black) and water oxygen–oligomer ether oxygen (red) for 0.02 mass fraction of water in the tetraethylene glycol (solid line) and in the PEG200 (dotted line) obtained with the water force fields SPC/E in combination with the OPLS forcefield.

**Figure 7 molecules-29-02070-f007:**
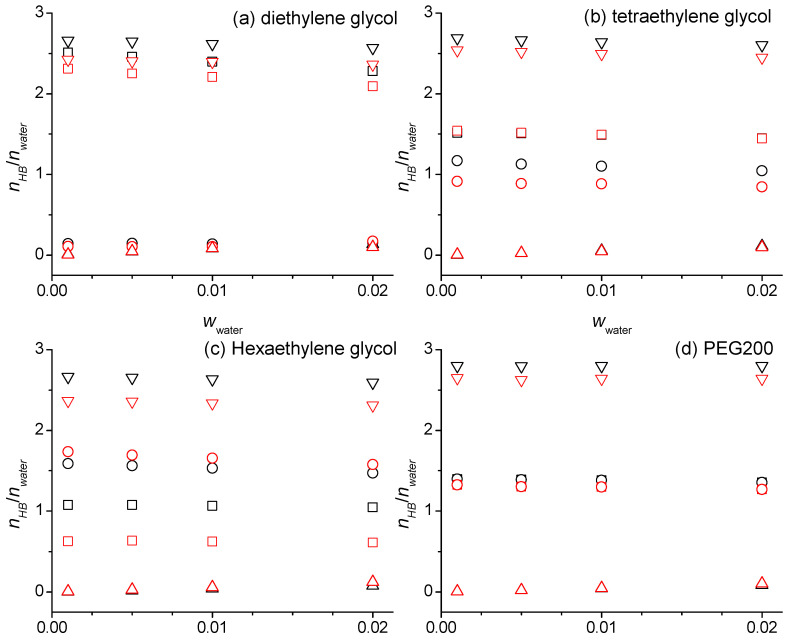
Number of the hydrogen bonds per number of the water molecules for the intermolecular hydrogen bonding of the water with hydroxy groups (squares), ether groups (circles), other water molecules (triangle up), and the total of all these water hydrogen bonding interactions (triangle down) as a function of the water mass fractions, *w*_water_, at 328 K for diethylene glycol (**a**), tetraethylene glycol (**b**), hexaethylene glycol (**c**), and PEG200 (**d**) obtained from the simulations using the SPC/E force field for the water and the OPLS (black symbols) or the modified OPLS force field (red symbols) for the oligomers.

**Figure 8 molecules-29-02070-f008:**
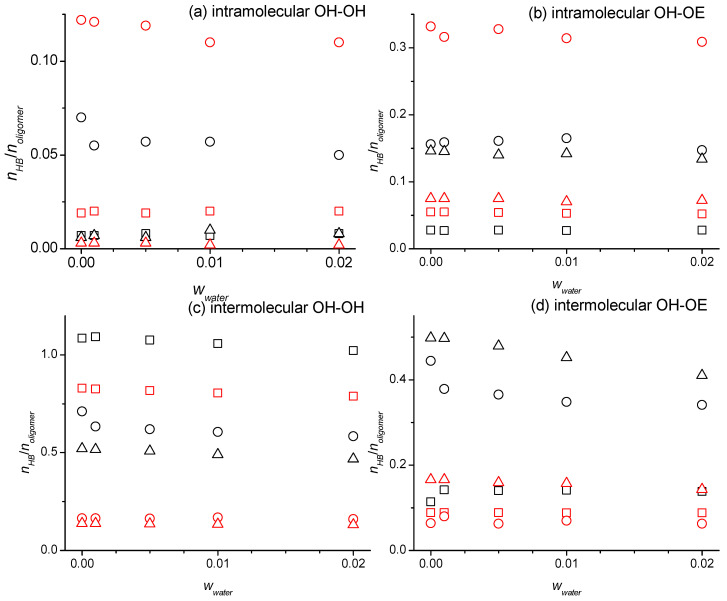
Number of hydrogen bonds per number of the oligomer molecules for intramolecular hydrogen bonding between (**a**) the two hydroxy groups and (**b**) the hydroxy and ether groups and for intermolecular hydrogen bonding between (**c**) the two hydroxy groups and (**d**) hydroxy and ether groups as a function of the water mass fractions, *w*_water_, at 328 K for the diethylene glycol (squares), tetraethylene glycol (circles), and hexaethylene glycol (triangle up), obtained from the simulations using the SPC/E force field for water and the OPLS (black symbols) or the modified OPLS force field (red symbols) for the oligomers.

**Figure 9 molecules-29-02070-f009:**
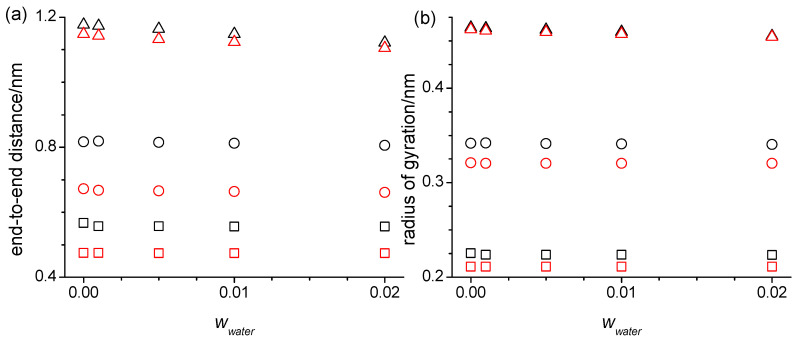
(**a**) End-to-end distances and (**b**) the radii of gyration as a function of the water mass fractions, *w*_water_, at 328 K for the diethylene glycol (squares), tetraethylene glycol (circles), and hexaethylene glycol, (triangle up), obtained from the simulations using the SPC/E force field for water and the OPLS (black symbols) or the modified OPLS force field (red symbols) for the oligomers.

**Figure 10 molecules-29-02070-f010:**
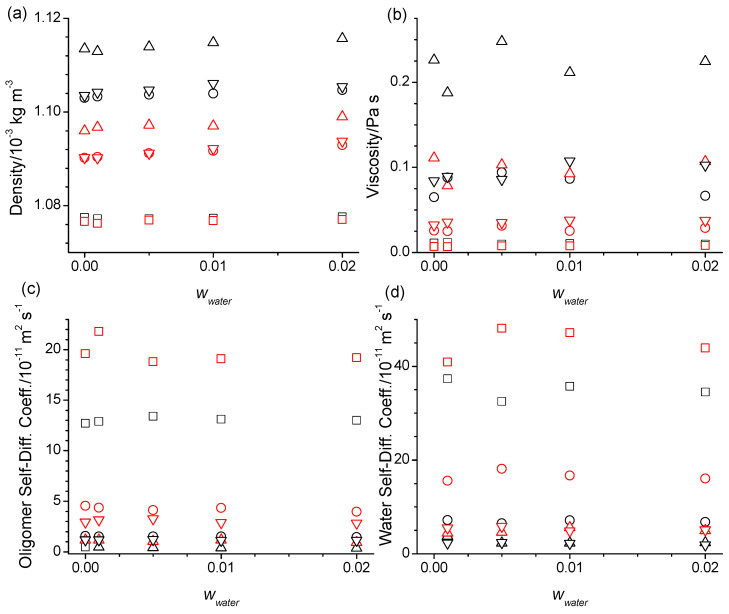
(**a**) Densities, (**b**) viscosities, and self-diffusion coefficients of the (**c**) oligomers and (**d**) water as a function of the water mass fractions, *w*_water_, at 328 K for the diethylene glycol (squares), tetraethylene glycol (circles), hexaethylene glycol (triangle up), and PEG 200 (triangle down) obtained from the simulations using the SPC/E force field for water and the OPLS (black symbols) or the modified OPLS force field (red symbols) for the oligomers.

**Table 1 molecules-29-02070-t001:** Simulation results of densities, viscosities, and self-diffusion coefficients at 328 K.

		*w_water_*					*w_water_*				
OPLS	Model ^a^	0 [16]	0.001	0.005	0.010	0.020	0 [16]	0.001	0.005	0.010	0.020
		Density/kg‧m^−3^					Viscosity/mPa‧s				
		Diethylene glycol					Diethylene glycol				
Exp. [15]		1091.7	1091.6	1091.4	1091.2	1090.6	9.0	9.0	9.0	8.9	8.7
Unmod	SPC/E	1077.5	1077.2	1077.2	1077.3	1077.6	11.2	11.8	9.8	10.6	9.9
Unmod	TIP4P	1077.5	1077.4	1077.7	1077.8	1078.0	11.2	10.5	10.4	11.2	11.2
Mod	SPC/E	1076.6	1076.2	1076.9	1076.8	1077.0	7.1	6.8	7.5	7.7	8.3
Mod	TIP4P	1076.6	1076.3	1076.5	1077.0	1077.8	7.1	7.0	7.3	7.7	7.8
		Tetraethylene glycol					Tetraethylene glycol				
Exp. [15]		1096.0	1096.0	1095.9	1095.9	1095.8	13.2	13.2	13.1	12.9	12.6
Unmod	SPC/E	1103.0	1103.3	1103.7	1103.9	1104.7	65.1	88.2	94.1	86.5	66.6
Unmod	TIP4P	1103.0	1103.2	1103.2	1103.6	1104.6	65.1	84.2	71.5	67.8	84.7
Mod	SPC/E	1090.2	1090.4	1091.2	1091.7	1092.9	25.6	25.1	31.3	25.4	28.9
Mod	TIP4P	1090.2	1090.3	1091.0	1091.8	1093.4	25.6	23.7	29.1	31.6	35.2
		Hexaethylene glycol					Hexaethylene glycol				
Exp. [15]		1099.8	1099.5	1098.1	1096.4	1092.9	18.3	18.2	17.9	17.6	16.9
Unmod	SPC/E	1113.5	1112.9	1113.9	1114.8	1115.7	226.0	187.4	247.7	211.2	224.2
Unmod	TIP4P	1113.5	1113.6	1114.3	1114.2	1115.8	226.0	248.5	238.5	251.9	258.8
Mod	SPC/E	1096.0	1096.7	1097.2	1097.0	1099.0	111.0	78.3	102.9	92.5	106.8
Mod	TIP4P	1096.0	1096.6	1096.9	1098.0	1099.2	111.0	107.8	108.1	83.8	120.0
		PEG200					PEG 200				
Exp. [14]		1097.0	1097.0	1097.0	1097.0	1097.0	14.7	14.7	14.6	14.5	14.3
Unmod	SPC/E	1103.5	1104.2	1104.7	1106.1	1105.4	84.4	89.4	85.9	107.4	102.5
Unmod	TIP4P	1103.5	1104.0	1104.5	1105.5	1106.2	84.4	78.2	89.8	89.0	110.7
Mod	SPC/E	1090.3	1090.2	1091.2	1092.2	1093.8	32.2	35.8	35.3	38.1	37.8
Mod	TIP4P	1090.3	1090.3	1091.3	1091.6	1093.7	32.2	40.2	36.8	44.6	38.4
		Oligomer self-diffusion/10^−11^ m^2^‧s^−1^					Water self-diffusion/10^−11^ m^2^‧s^−1^				
		Diethylene glycol					Diethylene glycol				
Exp. [15]		17.4	17.5	17.6	17.9	18.3					
Unmod	SPC/E	12.7	12.9	13.4	13.1	13.0		37.3	32.5	35.7	34.5
Unmod	TIP4P	12.7	12.8	12.8	13.2	12.4		27.8	29.8	31.0	37.0
Mod	SPC/E	19.6	21.8	18.8	19.1	19.2		40.9	48.1	47.2	43.9
Mod	TIP4P	19.6	20.8	21.0	18.6	18.5		44.0	47.2	41.3	40.2
		Tetraethylene glycol					Tetraethylene glycol				
Exp. [15]		10.1	10.1	10.2	10.4	10.6					
Unmod	SPC/E	1.56	1.51	1.52	1.52	1.45		7.21	6.52	7.17	6.79
Unmod	TIP4P	1.56	1.50	1.52	1.61	1.47		6.12	6.97	7.59	6.62
Mod	SPC/E	4.55	4.37	4.12	4.35	3.97		15.60	18.15	16.71	16.06
Mod	TIP4P	4.55	4.27	4.17	4.20	3.73		13.52	15.48	14.73	14.87
		Hexaethylene glycol					Hexaethylene glycol				
Exp. [15]		6.6	6.7	6.7	6.8	7.0					
Unmod	SPC/E	0.47	0.47	0.42	0.40	0.37		3.73	2.30	2.28	2.35
Unmod	TIP4P	0.47	0.44	0.42	0.39	0.36		2.59	2.22	2.15	2.19
Mod	SPC/E	1.15	1.17	1.03	1.15	0.92		4.41	4.58	5.61	4.98
Mod	TIP4P	1.15	1.06	1.08	1.06	0.96		3.40	4.33	4.26	4.56
		PEG200					PEG200				
Exp. [14]		8.20	8.24	8.41	8.62	9.04					
Unmod	SPC/E	1.20	1.19	1.22	1.13	1.14		2.28	2.44	2.24	1.91
Unmod	TIP4P	1.20	1.21	1.20	1.18	1.11		2.14	2.34	2.12	1.93
Mod	SPC/E	2.95	3.16	3.28	2.90	2.84		5.55	5.90	4.89	5.10
Mod	TIP4P	2.95	3.13	3.12	3.01	2.80		4.47	5.71	5.46	5.39

^a^ The specific TIP4P model used was TIP4P/2005.

**Table 2 molecules-29-02070-t002:** Simulation results of heat capacities, *C_p_*, and thermal expansion coefficients, *α*, at 328 K.

		*w_water_*					*w_water_*				
OPLS	Model ^a^	0 [16]	0.001	0.005	0.010	0.020	0 [16]	0.001	0.005	0.010	0.020
		*C_p_*/J‧mol^−1^‧K^−1^					*α*/1000 K^−1^				
		Diethylene glycol					Diethylene glycol				
Exp.		256.8 [29]					0.677 [15]				
Unmod	SPC/E	493.4	467.6	472.7	447.1	425.1	1.080	0.906	0.986	0.967	0.934
Unmod	TIP4P	493.4	476.0	468.8	439.3	430.0	1.080	1.026	0.975	0.912	0.909
Mod	SPC/E	507.7	469.6	464.2	446.5	434.0	1.044	1.037	0.980	1.015	1.073
Mod	TIP4P	507.7	472.2	459.1	432.8	426.2	1.044	1.093	1.064	0.912	0.966
		Tetraethylene glycol					Tetraethylene glycol				
Exp.		438.6 [29]					0.685 [15]				
Unmod	SPC/E	896.7	854.2	809.1	794.3	733.1	0.954	0.828	0.864	0.919	0.906
Unmod	TIP4P	896.7	878.7	905.2	792.9	718.2	0.954	0.988	1.328	0.869	0.876
Mod	SPC/E	818.6	826.6	789.3	744.9	713.7	0.956	0.896	0.859	0.905	0.934
Mod	TIP4P	818.6	800.3	799.6	770.5	700.7	0.956	0.862	0.845	0.954	0.936
		Hexaethylene glycol					Hexaethylene glycol				
Exp.		625.3 [29]					0.750 [15]				
Unmod	SPC/E	1313.4	1255.4	1109.9	1083.8	925.8	0.961	0.965	0.747	0.854	0.771
Unmod	TIP4P	1313.4	1172.7	1074.1	1091.9	915.5	0.961	0.675	0.764	0.932	0.885
Mod	SPC/E	1232.3	1182.6	1156.0	1110.1	940.8	0.928	0.910	0.891	0.950	0.840
Mod	TIP4P	1232.3	1199.1	1075.9	1071.0	977.0	0.928	0.906	0.792	0.850	1.036
		PEG200					PEG 200				
Exp.		431.0 [16]					0.727 [14]				
Unmod	SPC/E	916.7	952.8	901.9	848.1	820.9	0.932	0.902	0.896	0.913	0.903
Unmod	TIP4P	916.7	861.6	847.5	864.4	895.3	0.932	0.696	0.884	1.035	1.058
Mod	SPC/E	900.9	889.4	805.0	835.4	898.2	0.909	0.868	0.791	0.916	0.774
Mod	TIP4P	900.9	867.3	910.2	899.4	781.7	0.909	0.918	0.965	1.027	0.865

^a^ The specific TIP4P model used was TIP4P/2005.

## Data Availability

The original contributions presented in the study are included in the article and Appendix A, further inquiries can be directed to the corresponding authors.

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
