# Peer review of "Molecular Dynamics Study of the Green Solvent Polyethylene Glycol with Water Impurities"

_molecules, 2024, doi:10.3390/molecules29092070_

Round 1
Reviewer 1 Report
Comments and Suggestions for Authors
Hoffmanm et al. present an interesting computational investigation on the green solvent polyethylene glycol with water impurities. To some extent, the methods employed are quite routine, in my opinion. However, some interesting clues are in any case found and reported.
As stated by the Authors, they find an overall -- even though slight -- H-bond overstructuring due to the presence of water, in that the obtained water RDFs are essentially identical regardless of water content up to the maximum studied water amount. However, some nuanced effects on the H-bonding interactions are noticed, which in turn lead to varied water-dependencies for the densities, viscosities and self-diffusion coefficients. In this respect, I'm curious to understand which is the role of local electric fields in the H-bond structuring the Authors find. It has been recently shown, indeed, that local electric fields (like those present at the proximity of charged functional groups) are capable of strengthening the water H-bond network [Nat. Commun. 15, 1856 (2024)]. Comments on this important peculiar aspect would collocate the current work in a broader context and might render it more appealing.
Finally, the Authors report some tests on the OPLS force fields but not extensive tests were made on the water model. In fact, only two relatively simple force fields simulating the behavior of water are reported. Comments on this aspect would improve the work as well.
Once those aspects will be fixed it will be my pleasure to re-consider the current manuscript for publication.
Author Response
Author reply: To address both comments from Reviewer 1 we added the following text near the end of the Results and Discussion section.
We did not test additional other water force fields given that besides the water self-diffusion coefficient all other structural and physical properties showed water model independence. We should also point out that the OPLS and water force fields used in this study do not capture possible effects of local electric fields, which ab-initio MD simulations would capture [43]. However, these local field effects tend to increase structural organization which would only further increase hydrogen bonding interactions that appear to be overestimated by the force fields tested for the oligomers of PEG200 [16].
Reviewer 2 Report
Comments and Suggestions for Authors
The manuscript presents Molecular Dynamics simulations of PEG200 to study the effect of added water impurities. The simulation results show that the experimental observation that the addition of water hardly impacts the density, viscosity and self-diffusion of PEG.
The paper is well written and presents very interesting results in detail. The discussion of the results is exhaustive, and the conclusions refer to experimental work using PEG. My comments concern more the technical side of the manuscript.
Can Fig. 2 be presented in such a way that in point b) the reader can see more water molecules in the immediate vicinity of the glycol molecules, because only one is clearly visible; in c) can this part of the figure be enlarged to better see the mutual arrangement of the molecules?
On the other hand, I recommend reducing the size of the graphs in Fig. 3-Fig. 6, because they are disproportionately large compared to the rest of the article.
In my opinion, Tables S5 and S6 should be included in the main text of the work, because they contain a comparison to experimental values, which greatly enriches the work and shows how the simulation methods used refer to the experimental results.
After making these small corrections, the paper can be published.
Author Response
Can Fig. 2 be presented in such a way that in point b) the reader can see more water molecules in the immediate vicinity of the glycol molecules, because only one is clearly visible; in c) can this part of the figure be enlarged to better see the mutual arrangement of the molecules?
Author reply: We could certainly show more water molecules in Figure 2b, but this would be not helpful in this case because it would be distracting as this particular example shows already nicely all the pertinent water – ethylene glycol oligomer interactions that are needed to interpret the various peaks in the radial distribution functions shown in later figures. To increase the size of Figure c) and moving it below Figures a) and b) in the process is a good idea. This does not show yet in the submitted revised manuscript file because MDPI’s editorial staff can accomplish this task in a fraction of time than it would take us dealing with the template file.
On the other hand, I recommend reducing the size of the graphs in Fig. 3-Fig. 6, because they are disproportionately large compared to the rest of the article.
Author reply: We agree and again would like delegate this task shrinking the two figures in each case to fit side by side to the MDPI’s editorial staff.
In my opinion, Tables S5 and S6 should be included in the main text of the work, because they contain a comparison to experimental values, which greatly enriches the work and shows how the simulation methods used refer to the experimental results.
Author reply: We agree. We moved these tables into the manuscript and would like to ask the editorial staff to help with addressing the final formatting of these two tables.
Reviewer 3 Report
Comments and Suggestions for Authors
The paper entitled:” Molecular Dynamics Study of the Green Solvent Polyethylene Glycol With Water Impurities” by Markus M. Hoffmann, Matthew D. Too, Nathaniel A. Paddock, Robin Horstmann, Sebastian Kloth, Michael Vogel and Gerd Buntkowsky is a well-written a fully detailed manuscript which reports a molecular dynamics investigation aimed at assessing the influence of water impurities on physical properties such as density, viscosity and self-diffusion of Polyethyleneglycol.
In my opinion the manuscript deserves the publication on “Molecules”.
I have only very minor suggestions to the authors:
1) PAGE 5, LINES 155-157 The authors wrote:” The (HO)-C-C-O dihedral potential function was reduced by ½ for di-, tri, hexa- and heptaethylene glycol and by ¾ for tetra- and pen- taethylene glycol.” For completeness the reason of the reduction could be added to the sentence.
2) According to the templating guidelines, I think it is better to move the description of the used methodologies (Paragraph 2) in a new paragraph “4” called “Materials and Methods”.
Author Response
I have only very minor suggestions to the authors:
1) PAGE 5, LINES 155-157 The authors wrote:” The (HO)-C-C-O dihedral potential function was reduced by ½ for di-, tri, hexa- and heptaethylene glycol and by ¾ for tetra- and pen- taethylene glycol.” For completeness the reason of the reduction could be added to the sentence.
Author reply: The reason was mentioned in the introduction, these changes provided closest agreement of simulated densities, viscosities and self-diffusion coefficients. To better clarify, we changed the appropriated sentence in Section 2 to:
Further specific details including all parameters associated with the OPLS force field are listed in completeness in the supplementary materials of prior work [16] and need not be repeated here except to summarize the adjustments of the modified OPLS forcefield which resulted in the closest agreement between simulated and experimental physical properties and are as follows (see also Figure 1):
2) According to the templating guidelines, I think it is better to move the description of the used methodologies (Paragraph 2) in a new paragraph “4” called “Materials and Methods”.
Author reply: We refrained from the title of “Materials and Methods” as the reported study is completely computational and thus entitled the section header to “Computational Methods”. It appears that the editorial staff of MDPI is agreeable to this section title.
Reviewer 4 Report
Comments and Suggestions for Authors
Please check the file

Author Response
- The key word is "polyethylene glycol; ethylene glycol oligomers; MD simulations;
hydrogen bonding; radial distribution functions; density; self-diffusion; viscosity", some
words are too broad.
Author reply: We believe including broad entries ensures that the manuscript will show up during literature searches by researchers. Nevertheless, we removed “MD simulations” and put instead the phrase “water impurity” given that none of the other keywords mention water, which is a key subject of this study.
- In line 261, the caption of Figure 2 appears to be wrong. Are b) and c) reversed for “b) just
the water molecules, and c) a water molecule interacting with hexamethylene glycol
(oxygen = red, hydrogen = white, carbon = teal.).”
Author reply: We thank the reviewer for catching this oversight stemming from changing the picture order before submitting the manuscript but failing to swap the descriptors in the caption as well. In the process we caught that one longer paragraph was also not updated and had still incorrect figure references, which we corrected.
Minor:
- In lines 158-160, “The parameters for the 3-point SPC/E and the 4-point TIP4P/2005 water
model are included in the GROMACS 2020.4 package and were used without any
modifications.” “Were” should be “are”.
Author reply: Grammatically, we think that this change would be incorrect as the work is completed and the models were (not “are”) not changed for the completed study.
- In lines 45-46, “First reviews on the initial successes of PEG in chemical synthesis
appeared in 2005[2,3].” It is recommended to add a blank between ‘2005’ and ‘[2,3]’. And
same issue on lines 85-86, “…were evaluated[16].” Please check the whole manuscript.
- In line 143, the expression ‘Cn’ in “with Cn torsional energy barrier coefficients for …”
does not seem to be a subscript, which different with its representation in the equation (1).
- In line 176, ‘kJ mol-1 ps-1’; in line 57, ‘200 g∙ mol-1’; in line 610, ‘200 g/mol’. It is
recommended to keep consistent in full manuscript
Author reply: These editorial corrections were all completed.
Round 2
Reviewer 1 Report
Comments and Suggestions for Authors
The paper can now be accepted.